# Routine CT Diagnostics Cause Dose-Dependent Gene Expression Changes in Peripheral Blood Cells

**DOI:** 10.3390/ijms26073185

**Published:** 2025-03-29

**Authors:** Hanns Leonhard Kaatsch, Laura Kubitscheck, Simon Wagner, Thomas Hantke, Maximilian Preiss, Patrick Ostheim, Tim Nestler, Joel Piechotka, Daniel Overhoff, Marc A. Brockmann, Stephan Waldeck, Matthias Port, Reinhard Ullmann, Benjamin V. Becker

**Affiliations:** 1Department of Radiology and Neuroradiology, Bundeswehr Central Hospital, 56072 Koblenz, Germany; hannsleonhardkaatsch@bundeswehr.org (H.L.K.);; 2Bundeswehr Institute of Radiobiology affiliated to Ulm University, 80937 Munich, Germany; 3Department of Urology, Bundeswehr Central Hospital, 56072 Koblenz, Germany; 4Department of Neuroradiology, University Medical Center Mainz, 55131 Mainz, Germany

**Keywords:** computed tomography, low-dose radiation, risk assessment, gene expression, whole transcriptome sequencing

## Abstract

The increasing use of computed tomography (CT) has led to a rise in cumulative radiation dose due to medical imaging, raising concerns about potential long-term adverse effects. Large-scale epidemiological studies indicate a higher tumor incidence associated with CT examinations, but the underlying biological mechanisms remain largely unexplained. To gain further insights into the cellular response triggered by routine CT diagnostics, we investigated CT-induced changes of gene expression in peripheral blood cells using whole transcriptome sequencing. RNA was isolated from peripheral blood cells of 40 male patients with asymptomatic microhematuria, sampled before and after multi-phase abdominal CT (CTDIvol: 3.75–26.95 mGy, median: 6.55 mGy). On average, 22.11 million sequence reads (SD 5.71) per sample were generated to identify differentially expressed genes 6 h post-exposure by means of DESeq2. To assess the dose dependency of CT-induced effects, we additionally divided samples into three categories: low exposure (≤6.55 mGy, n = 20), medium exposure (>6.55 mGy and <12 mGy, n = 16), and high exposure (≥12 mGy, n = 4), and repeated gene expression analysis for each subset and their corresponding prae-exposure sample. CT exposure caused consistent and dose-dependent upregulation of six genes (*EDA2R*, *AEN*, *FDXR*, *DDB2*, *PHLDA3*, and *MIR34AHG*; padj < 0.1). These genes share several functional commonalities, including regulation by TP53 and involvement in the DNA damage response. The biological pathways highlighted by Gene Set Enrichment Analysis (GSEA) suggest a dose-dependent increase of cellular damage and metabolic particularities in the low-exposure subset, which may be related to a potential adaptive cellular response to low-dose irradiation. Irrespective of applied dose, *AEN* emerged as the most robust biomarker for CT exposure among all genes. Routine abdominal CT scans cause dose-dependent gene deregulation in association with DNA damage in peripheral blood cells after in vivo exposure. Regarding risk assessment of CT, our results support the commonly applied “As Low–As –Reasonably Achievable (ALARA)” principle. Evidence of additional gene expression changes associated with metabolic processes indicates a rather complex molecular response beyond DNA damage after CT exposure, and emphasizes the need for further targeted investigations.

## 1. Introduction

Computed tomography (CT) is a powerful diagnostic tool whose medical value is undisputed. However, as a consequence of its broad and frequent application of CT, the population-wide administered dose has continuously increased in recent decades [1,2]. In response to this development, various technological improvements and organizational measures have been implemented to reduce the administered dose as much as reasonably achievable [3,4]. In the U.S., these efforts have even reversed the per capita dose trend in recent years [5]. Yet, CT still accounts for the largest share of collective effective dose among medical imaging modalities [6,7,8], with each CT examination presenting an X-ray exposure with potential harmful effects on the patient.

Epidemiological quantification of the individual risk for adverse long-term effects from CT exposure is challenging. Large cohort sizes are necessary to gain sufficient statistical power to capture the subtle effects expected after CT exposures. A recent multinational study on the incidence of malignancies in 876,771 young individuals subjected to CT examinations reported an excess relative risk of 1.96 per 100 mGy for hematological malignancies, and to a similar extent for lymphoid and myeloid tumors [9]. This observation is consistent with earlier epidemiologic evidence of an increased risk of CT-induced malignancies, especially in younger patients and after repeated exposures [10,11,12,13,14,15,16].

Radiobiological investigations can complement epidemiology-based risk stratification by highlighting the cellular processes underlying irradiation-associated long-term effects and providing further clues on the dependency of these effects on dose, CT protocols, and other factors. In previous studies based on ex vivo irradiated peripheral blood cells, we have evaluated the effect of CT exposure on gene expression, epigenetic features, and the frequency of DNA double-strand breaks, including both dual-energy and photon-counting CT [17,18,19,20]. Despite the use of state-of-the-art devices with modern dose-reduction techniques, our studies reproducibly demonstrated an impact of modern CT diagnostics on DNA integrity and gene expression, which is in line with corresponding findings from other studies on CT-associated biological effects in blood cells [21,22,23].

The experimental design of our previous studies, taking advantage of the controlled settings of ex vivo exposure of peripheral blood cells drawn from healthy volunteers, aimed to minimize possible confounding factors that could potentially mask irradiation-associated effects. In the present study, we addressed the question of how far our ex vivo findings translate into the clinical setting of routine CT examinations. For this purpose, we employed whole transcriptome sequencing to compare gene expression profiles of peripheral blood cells sampled from 40 patients with asymptomatic microhematuria before and after multi-phase abdominal CT.

## 2. Results

### 2.1. Identification of Genes Differentially Expressed Before and After CT Exposure

A total of 80 libraries, representing 40 pairs of corresponding prae- and post-exposure samples, were successfully sequenced, with an average read number of 21.11 million reads (SD 5.71). Exploratory analysis using hierarchical clustering of sample-to-sample distances was performed to reveal overall similarities in gene expression profiles. For all patients, hierarchical clustering based on Euclidian distance resulted in pairwise clustering of corresponding prae- and post-exposure samples (Appendix A). This corroborates our previous observations that interindividual gene expression characteristics, and not CT-induced effects, dominate overall gene expression signatures. Neither age characteristics of patients nor CT-associated features were enriched in the clusters of the sample-to-sample distance matrices. Differential gene expression analysis of prae- and post-exposure samples yielded seven genes—*EDA2R*, *AEN*, *FDXR*, *DDB2*, *PHLDA3*, *MIR34AHG*, and *RYR3*—to be significantly upregulated after CT exposure (padj < 0.1; Figure 1a). We did not detect any gene which was significantly downregulated in response to CT exposure. Among the seven upregulated genes, *RYR3* stands out. Unlike the other six genes, which showed a rather homogenous trend of upregulation across our patient group (see below), the significance of *RYR3* is borderline (padj = 0.099) and is solely owed to high expression in four patients, with considerable CT-induced upregulation in three of them (Figure 1b). A re-evaluation of medical records did not provide any clues on possible particularities of these patients in terms of confounding diseases or medication.

### 2.2. Analysis of Dose-Dependent Effects

To investigate the dependency of gene expression changes on administered dose, we divided the patient group into three subsets as described above (low, medium, and high exposure groups). Apart from *RYR3*, all candidate genes showed dose dependency when comparing the amount of normalized read counts per gene between the prae-exposure, low, medium, and high exposure groups, respectively (Figure 2). Dose dependency was most prominent for *FDXR* and *EDA2R*. One limitation of the comparison above is that it only considers the overall distributions of gene dose within each subset and not CT-associated increase of gene dose in each single patient. Therefore, we additionally calculated the CT-induced upregulation of *EDA2R*, *AEN*, *FDXR*, *DDB2*, *PHLDA3*, and *MIR34AHG* as the ratio of normalized read counts per gene before and after exposure for each individual patient. As depicted in the boxplots provided in Appendix A, patient-specific upregulation was dose-dependent for all six genes. The observation of dose-dependent upregulation raises the question of the reliability of these genes as biomarkers across different CT doses. For example, does a moderate increase of a given gene in the low-exposure group suffice to reach significance? In order to address this question, we repeated differential gene expression analysis by means of DESeq2 separately for each subset (low, medium, and high exposure), with each corresponding prae-exposure sample as a reference. The Upset plot presented in Figure 3 summarizes all genes found to be significant (padj < 0.1) and their overlap between the different exposure groups. As inferred from this plot, *AEN* is the gene that reaches significance in all analysis and hence presents as the most stable biomarker for CT exposure in this study. This even applies to the high-exposure group, where the very small number of patients limits statistical power and is unlikely capable of equalizing the profound effects of interindividual genetic heterogeneity, which may mask CT-associated effects.

### 2.3. Analysis of Gene Interactions and Biological Pathways

Up to now, our analysis considered each gene as a separate entity and disregarded any functional relationship. To investigate possible biological interactions between the differentially expressed genes identified in this study, we uploaded the six protein-coding genes (excluding *MIR34AHG*, a long non-coding RNA) to the STRING database (version 12.0) hosted at https://string-db.org/ (accessed on 19 January 2025) [24]. As depicted in Figure 4, EDA2R, AEN, FDXR, DDB2, and PHLDA3 showed interactions with medium to high confidence, where confidence refers to a combined score calculated by STRING based on co-expression and co-mentioning in the literature. Again, RYR3 stands out, as it remains an orphan within the network, even when expanding the network by first and second neighbors. According to STRING, the six proteins showed significant enrichment for Protein-Protein-Interaction (PPI enrichment *p*-value: 5.68 × 10^−7;^ *p*-value = 1.18 × 10^−7^ when excluding RYR3) and the biological process “Intrinsic apoptotic signaling pathway by p53 class mediator” (GO: 0072332; padj = 0.0040). To further explore the involvement of biological pathways in the cellular response to CT exposure, we employed preranked GSEA [25] based on 50 gene sets gathered in the MSigDB hallmark gene set collection [26] and a custom gene set comprising our six candidate genes. GSEA was performed for all DESeq2 results obtained in this study, including the dose-group-specific comparisons. As for the custom gene set, GSEA was highly significant across all subsets and the complete study group (padj < 0.01), with normalized enrichment scores of 2.135 (low), 2.088 (medium), 2.105 (high), and 2.187 (all samples). The heatmap shown in Figure 5 summarizes the 26 MSigDB hallmark gene sets that reached significance (padj < 0.1) in at least one of the exposure groups or in the comparison of all patients before and after CT examination.

## 3. Discussion

Here, we present the results of our study on transcriptomic changes in peripheral blood cells in response to routine CT diagnostics, which, to the best of our knowledge, represents the largest CT study of this kind to date. With the identification of significant upregulation of *AEN*, *EDA2R*, *DDB2*, and *FDXR*, we were able to replicate all four candidate genes from our previous ex vivo studies, suggesting that—at the point of time investigated—ex vivo CT exposure of peripheral blood adequately reflects the cellular response to CT exposure in vivo in a clinical setting.

In addition to the four genes mentioned above, three further genes showed significant associations with CT exposure in our study: *PHLDA3*, *MIR34AHG*, and *RYR3*. While CT-induced upregulation of *PHLDA3* and *MIR34AHG* was observed in a great proportion of our study group, the significance of *RYR3* was mainly attributed to the extreme upregulation in three patients. Thus, we do not consider *RYR3* a general biomarker for CT exposure but rather see it as an example for the potential of confounding factors to modulate the biological consequences of CT examinations. Unfortunately, we have failed to identify any particularities characterizing these patients, such as medication, co-existing illness, age, or day/time of examination.

Apart from *RYR3*, the degree of upregulation of CT-responsive genes was characterized by clear dose dependency. Dose-dependency of gene expression has already been reported in the literature [27,28,29], but robust and reproducible whole-transcriptome data in the very low-dose range as administered in this study are rare. In terms of clinical relevance, the observed dose dependency underscores the necessity to optimally trade off the requirements of image quality against risk avoidance even at such low doses, on the one hand as well as the strict indication for CT scans in general on the other hand. In the context of radiobiology, our observation of dose dependency in the very low-dose range supports the linear no-threshold model, which suggests that biological effects at low doses can be linearly extrapolated from observations at higher doses without a dose threshold for biological effects [30].

Investigating the biological effects of CT examinations not only assists in a better understanding of long-term risks associated with this diagnostic procedure but also provides a rare opportunity to study the effects of low-dose X-ray exposure in vivo in humans. This offers the chance to identify relevant biological processes and define new biomarkers for low-dose exposure. All six candidate genes of this study have already been shown to be inducible by irradiation-associated DNA damage at higher doses, with some already proposed as biomarkers of irradiation [27,31,32,33,34,35]. A further commonality of the six genes is their known regulation by TP53, which is in line with previous observations at higher doses [36,37,38,39]. Co-expression and literature data suggest that the six candidate genes may be functionally related (Figure 4). For example, all six candidate genes rank among the top 26 of the top 100 genes co-expressed with BAX, which encodes a TP53-controlled regulator of apoptosis [40]. Notably, *BAX* was one of five genes deregulated six hours after CT exposure in one of our previous ex vivo studies [17]. Given this background, one is tempted to speculate that upregulation of these six candidate genes does not represent independent events but rather that these genes act in concert in a common biological process triggered by low-dose exposure.

Given the rather low doses used in CT examinations in a radiobiological context, only subtle biological effects can be expected. Accordingly, clustering of samples based on similarities of their whole transcriptome profiles was determined by the interindividual differences in overall gene expression profiles and not CT induced effects. This implicates the risk that CT-induced effects may be masked by individual genetic background and possible heterogeneity of response to irradiation within the study group, thereby reducing statistical power dependent on dose as well as composition and size of the study group. Indeed, dose-dependent sub-setting of patients into three groups revealed that out of the six candidate genes, only *AEN* reached statistical significance in differential gene expression analyses across all subsets and the complete sample set (Figure 3). While this highlights the reliability of *AEN* as a biomarker for low-dose exposure, it also illustrates the statistical limitations of single-gene-focused approaches in capturing the subtle alterations of gene expression associated with low-dose exposure. Recently, it has been proposed to employ irradiation-specific gene expression signatures, in the sense of a metagene, for the characterization of irradiation-induced biological effects [41]. In this study, we combined all six candidate genes into a custom gene set that was subsequently used for GSEA. Despite five out of six genes not exceeding the statistical thresholds in differential gene expression analysis in all subsets, GSEA using our custom gene set demonstrated highly significant effects across all subsets and the complete sample group, with comparable normalized enrichment scores. Yet, the small number of genes in our custom gene set is a clear limitation, and future studies would benefit from gene sets larger than 8–10 genes [42]. As a final step in our analysis, we performed GSEA utilizing the MSigDB Hallmark gene sets. These public gene sets have been composed with the goal of representing a broad spectrum of well-defined biological states with coherent expression and as little redundancy as possible [26]. As depicted in Figure 5, GSEA results for the high-exposure group were characterized by an increase in significance and normalized enrichment scores for gene sets associated with DNA damage response and cell death pathways (p53, TNF-alpha signaling via NF-κB, apoptosis, Wnt-beta Catenin Signaling and UV response up), likely reflecting the more pronounced adverse biological effects with increasing CT dose. In addition, GSEA points to a prominent role for genes associated with oxidative phosphorylation in the low exposure and all sample groups. This is of special interest in the context of CT risk assessment, particularly regarding the effects of repeated CT, as oxidative phosphorylation has been reported as a beneficial adaptive response to DNA damage [43]. Yet, these observations warrant further verification, especially given the small size of the high exposure group and the unclear role of other processes such as Hypoxia, Glycolysis, and PI3K/AKT/mTOR pathways, whose upregulation by low-dose radiation has previously been linked to a potential beneficial adaptive outcome [44,45], but were enriched in the downregulated fraction of genes in the low exposure group in our analysis.

In this study, we were able to demonstrate that low-dose irradiation in the course of routine CT examination elicits a dose-dependent deregulation of genes involved in DNA damage and cellular stress response. Future investigations could focus on issues not addressed in this study, including the impact of contrast agents [46], CT protocols, examined body regions, and sex on the cellular response to CT exposure.

## 4. Materials and Methods

The data supporting the findings of this study are available on request from the corresponding authors (R.U./B.V.B.).

### 4.1. Study Population

Our study included 40 male patients (mean age: 58.88 ± 10.45 years), who were referred to our institution for the investigation of microhematuria between November 2019 and November 2023. This patient group was generally characterized by a homogeneous composition and few previous illnesses. To mitigate the influence of confounding factors on gene expression profiles, patients with known tumors or active inflammatory diseases, a history of recently undergone surgery, prior chemo/radiation therapy, or pathological CT findings were excluded from the study. This study was approved by the Medical Association of Rhineland Palatinate, Germany (reference number: 837.084.17(10918)).

### 4.2. CT Examination

All patients underwent a multiphase CT scan of the abdomen, consisting of a non-contrast-enhanced scan followed by contrast-enhanced scans in one of the following: the arterial phase of the upper abdomen (30 s post-injection), portal venous (90 s post-injection), post-contrast excretory phase of the whole abdomen (15 min post-injection; “phase_4”, n = 22), solely native and post-contrast excretory phase (“phase_2”, n = 1), or portal venous and post-contrast excretory phase of the whole abdomen (“phase_3”, n = 17). The CT examinations were performed on state-of-the-art third-generation CT scanners (Toshiba/Canon Aquilion One, Canon Medical Systems, Otawara, Japan, (n = 9); and Siemens SOMATOM go.TOP, Siemens Healthineers, Forchheim, Germany, (n = 31), in a randomized fashion depending on local device availability. For contrast-enhanced CT scans, 100 mL of Xenetix 300/350 were administered. CT scans and blood collection were only performed in the morning and early noon to counteract the possible influence of circadian rhythms on gene expression patterns.

### 4.3. Dose Calculations

The dose calculation was performed using Radimetrics™ (Version 3.4.6, Bayer Vital GmbH, Leverkusen, Germany). Dose data were extracted from PACS (JiveX, Visus Health IT GmbH, Bochum, Germany) into Radimetrics via dicom-tags or Radiation Dose Structured Reports (RDSR). The monte carlo phantoms used for dose simulation for the selected patient group were then individually adapted to the patients’ gender, stature, and weight. The conformity of the dose modulation curves with the actual scan area was checked by a medical physics expert and adjusted manually in the event of deviations. This was followed by Monte Carlo simulation of the organ and effective doses. All scan parameters (CTDI_vol_body, high-voltage tube current, and simulated organ and effective doses) were exported. For the statistical classification of the patient groups, dose values per slice applied during the CT-scan were considered in relation to the 32 cm body phantom (CTDI_vol_body).

### 4.4. Sample Acquisition, RNA-Isolation, and Whole-Genome RNA Sequencing

Peripheral blood samples were collected in EDTA-Monovettes (2.6 mL; Sarstedt, Nümbrecht, Germany) before and after CT scans. After blood draw, sample tubes were immediately incubated for 6 h at 37 °C. Afterwards, blood samples were transferred into PAXGene tubes (PreAnalytiX, Hombrechtikon, Switzerland), incubated for ~2 h at room temperature, and finally stored at −20 °C until further processing. Total RNA was isolated using the PAXgene Blood RNA Kit (PreAnalytiX, Hombrechtikon, Switzerland). Agilent 2100 Bioanalyzer System (Agilent, Santa Clara, CA, USA) was used for RNA quantification and quality assessment. RNA-Seq library preparation was conducted for all samples employing the NEBNext Ultra II Directional RNA Library Kit for Illumina following the manufacturer’s recommendations (NEBNext Ultra™ II Directional RNA Library Prep Kit for Illumina and rRNA depletion option, New England Biolabs Inc., Ipswich, MA, USA, protocol version 1.0, 4/17). After quantification with a Qubit 3.0 Fluorometer (Thermo Scientific, Waltham, MA, USA) and quality control by means of Agilent’s TapeStation system (Agilent, Santa Clara, CA, USA), samples were sequenced on the Illumina NextSeq500 sequencing platform (single-end; 1 × 75 bp; high-output reagent kit, Illumina, San Diego, CA, USA).

### 4.5. Differential Gene Expression Analysis

Illumina Nextseq500 sequencing output was demultiplexed and converted to FASTQ format using Illumina’s proprietary software tool bcl2fastq (version 2.19.0). Quality of sequencing reads was checked using FASTQC [47]. Transcript expression was quantified using the mapping-mode of Salmon (version 1.10.3) [48]. In brief, a Salmon-specific index with decoy sequences was generated on the basis of GRCh38_primary_assembly.genome.fa and gencode.v45.transcripts.fa, which was then used to map reads using the salmon quant command with default settings, including the gcBias and validateMappings flags. Salmon output was imported with TxiMeta, summarizing transcript-level read counts to gene level and differential gene expression analysis using DESeq2 was performed, following a workflow published by Love et al. [49,50]. Patient ID was included as a cofactor in the design formula to account for the paired design of our study. In a first step of differential gene expression analysis, we contrasted all samples taken before and after CT exposure and defined the differentially expressed genes (padj < 0.1). In addition to this gene focused approach, we further explored the possible involvement of specific biological pathways in the cellular response to CT exposure. For this purpose, the Python (version 3.12.2) package GSEA.py (version 1.1.4) [51] and the MSigDB_Hallmark_2020 gene set [26] were utilized for GSEA of DESeq2 results. In this case, genes were not filtered by padj, but ranked with respect to the “stat” column. The “stat” value represents the quotient of log-fold changes and the corresponding log-fold standard error. GSEA takes the ranked results from DESeq2 analysis (or any other differential gene expression analysis) and tests for statistical relevant bias in the distribution of predefined gene sets within the table of ranked results. In this way, GSEA avoids the risk of losing relevant information by prefiltering results based on *p*-values or fold changes while enhancing statistical power by looking at gene sets instead of each gene separately.

For bioinformatics analysis of possible dose-dependent effects on gene expression profiles, we split samples taken after CT exposure into three groups, thresholded by the median CTDIvol (Computed Tomography dose index) administered in this study (6.55 mGy) and a national diagnostic reference value for abdominal CT examinations of 12 mGy [52], respectively. This resulted in: low exposure (≤6.55 mGy; n = 20), medium exposure (>6.55 mGy and <12 mGy; n = 16), and high exposure groups (≥12 mGy; n = 4). Based on these subsets, we repeated the differential gene expression analysis as described above using prae-exposure samples corresponding to each subset as a reference. Due to the small sample size of the high exposure group (n = 4), the design formula of the generalized linear model used for analysis of the high exposure subset did not include patient ID. For data visualization, ggplot2 and EnhancedVolcano package were used [53,54].

## Figures and Tables

**Figure 1 ijms-26-03185-f001:**
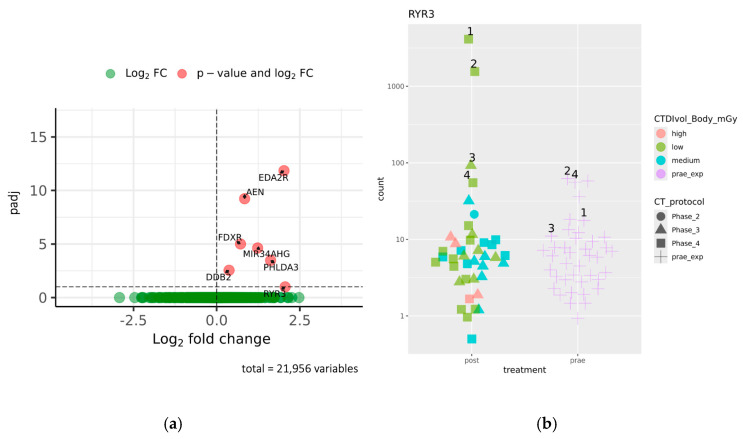
(**a**) Volcano plot highlighting genes differentially regulated after CT exposure. Padj values of all genes are plotted against their log fold change. The cutoff for statistical significance (−Log10 padj) is indicated by a horizontal dashed line. Significant genes are additionally named and distinguished by red color. (**b**) Plot of normalized gene counts for *RYR3* of each patient before and after CT exposure. The shape and color of each data point denote CT protocol and exposure group, respectively. Numbers next to selected samples mark the four patients with highest expression of *RYR3* post-exposure and their corresponding prae-exposure sample.

**Figure 2 ijms-26-03185-f002:**
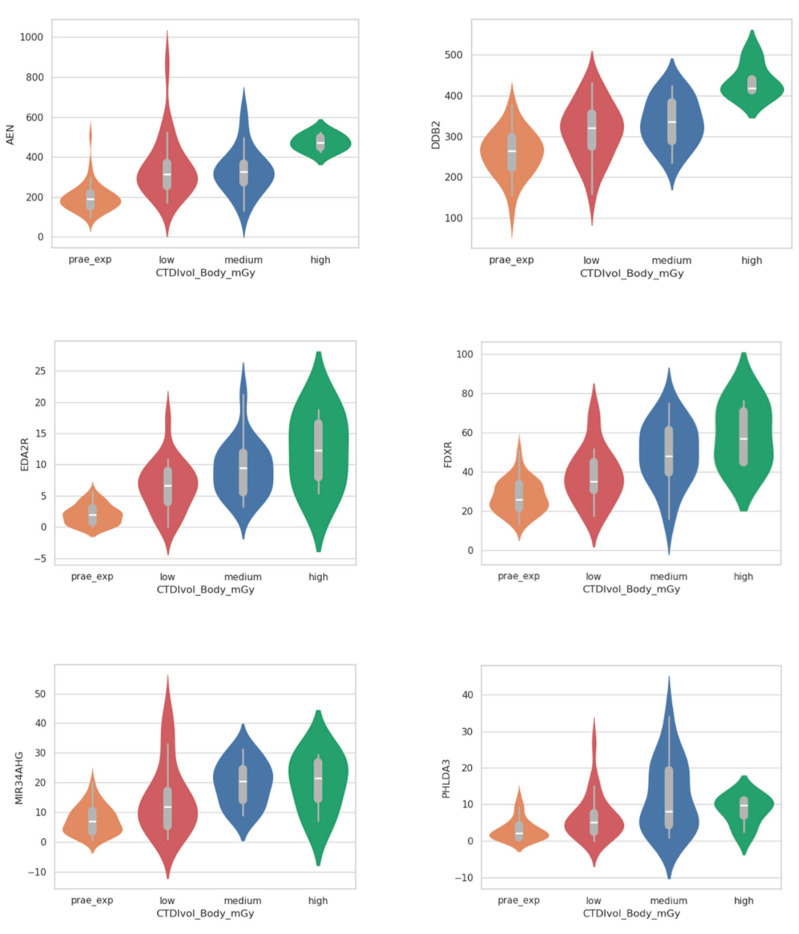
Violin plot with included boxplot depicting the distribution of normalized read counts of all six candidate genes for all patients, grouped by the exposure groups prae-exp, low, medium, and high.

**Figure 3 ijms-26-03185-f003:**
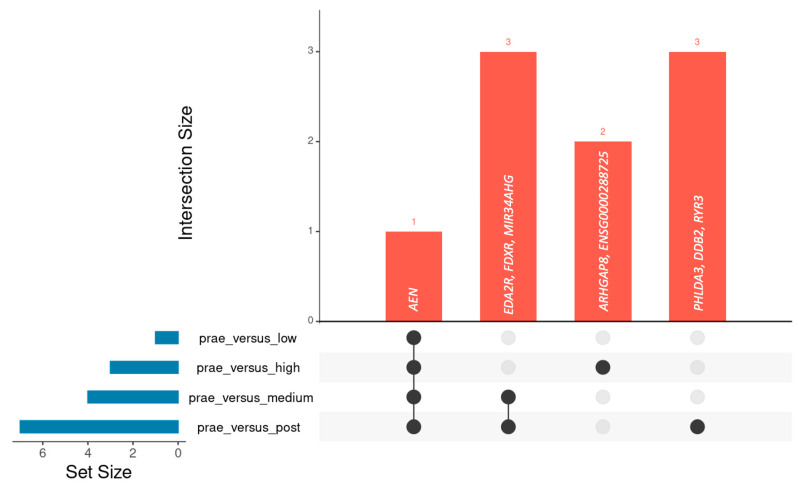
Upset plot summarizing the overlap of significantly deregulated genes between exposure groups. The histogram on the bottom left-hand side represents the total number of deregulated genes per exposure group. The bar plot on the top right-hand side represents the number and names of deregulated genes that shared upregulation among exposure groups, as depicted by the black circle(s) and lines in the matrix below.

**Figure 4 ijms-26-03185-f004:**
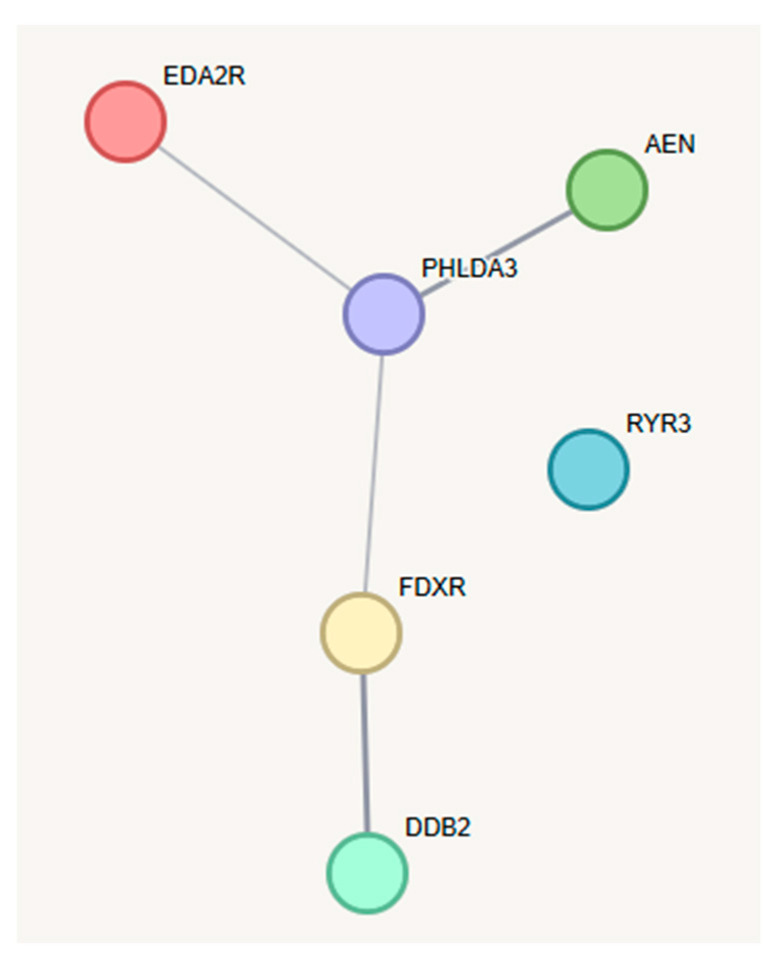
Protein-Protein Interaction Network: The STRING database was employed to depict interactions between the six protein-coding genes found significantly deregulated in this study. The strength of data support (medium and high confidence levels) is indicated by the thickness of edges (see Section 2 for details).

**Figure 5 ijms-26-03185-f005:**
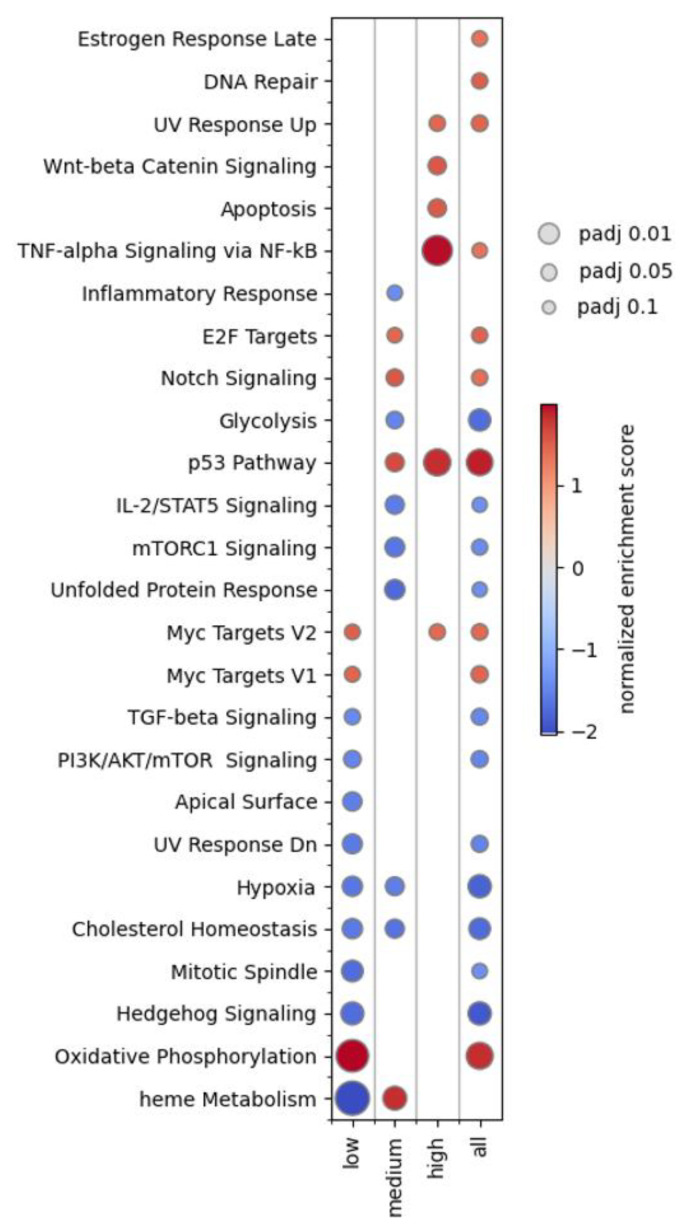
Heatmap summarizing GSEA results for all MSigDB Hallmark gene sets that reached significance in at least one of the exposure groups (low, medium, and high) or the all-sample group. Bubble size represents the significance level. Normalized enrichment scores are visualized by color grading. Positive and negative scores indicate enrichment in the fraction of upregulated and downregulated genes, respectively.

## Data Availability

The data that support the findings of this study are available upon request from the corresponding authors (R.U./B.V.B.).

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
