# Peer review of "Routine CT Diagnostics Cause Dose-Dependent Gene Expression Changes in Peripheral Blood Cells"

_ijms, 2025, doi:10.3390/ijms26073185_

Round 1
Reviewer 1 Report
Comments and Suggestions for Authors
The paper investigates the dose-dependent effects of routine CT diagnostics on gene expression in peripheral blood cells. Using whole transcriptome sequencing, it identifies specific genes that are upregulated post-CT exposure, with potential implications for radiation risk assessment.
The major novelties of the article are:
1. Demonstration of Dose-Dependent Gene Expression Changes. The study provides strong evidence that CT exposure causes dose-dependent upregulation of six genes (EDA2R, AEN, FDXR, DDB2, PHLDA3, MIR34AHG), primarily linked to DNA damage response and TP53 regulation. Among them, AEN is identified as the most stable biomarker.
2. Use of Whole Transcriptome Sequencing in a Clinical Setting. Unlike previous ex vivo studies, this study analyzes in vivo gene expression changes post-CT, improving the clinical relevance of the findings.
3. Biological Pathway Analysis Suggesting Adaptive Cellular Responses. Gene Set Enrichment Analysis (GSEA) reveals potential adaptive metabolic responses at low radiation doses, particularly in oxidative phosphorylation pathways.
4. Validation of Prior Ex Vivo Findings in a Clinical Context. The study replicates previous findings regarding the upregulation of key radiation-responsive genes, strengthening confidence in their role as biomarkers.
5. Implications for the Linear No-Threshold (LNT) Radiation Model. The observed dose-dependent gene expression supports the LNT model, suggesting that even low-dose radiation triggers measurable biological responses.
The study presents novel and clinically relevant insights into CT-induced gene expression changes.
Author Response
Reviewer 1:
The paper investigates the dose-dependent effects of routine CT diagnostics on gene expression in peripheral blood cells. Using whole transcriptome sequencing, it identifies specific genes that are upregulated post-CT exposure, with potential implications for radiation risk assessment.
The major novelties of the article are:
1. Demonstration of Dose-Dependent Gene Expression Changes. The study provides strong evidence that CT exposure causes dose-dependent upregulation of six genes (EDA2R, AEN, FDXR, DDB2, PHLDA3, MIR34AHG), primarily linked to DNA damage response and TP53 regulation. Among them, AEN is identified as the most stable biomarker.
2. Use of Whole Transcriptome Sequencing in a Clinical Setting. Unlike previous ex vivo studies, this study analyzes in vivo gene expression changes post-CT, improving the clinical relevance of the findings.
3. Biological Pathway Analysis Suggesting Adaptive Cellular Responses. Gene Set Enrichment Analysis (GSEA) reveals potential adaptive metabolic responses at low radiation doses, particularly in oxidative phosphorylation pathways.
4. Validation of Prior Ex Vivo Findings in a Clinical Context. The study replicates previous findings regarding the upregulation of key radiation-responsive genes, strengthening confidence in their role as biomarkers.
5. Implications for the Linear No-Threshold (LNT) Radiation Model. The observed dose-dependent gene expression supports the LNT model, suggesting that even low-dose radiation triggers measurable biological responses.
The study presents novel and clinically relevant insights into CT-induced gene expression changes.
Thank you for your review and for highlighting the key aspects of our study. We are glad that our findings resonate with you and we appreciate your positive remarks.
Reviewer 2 Report
Comments and Suggestions for Authors
IJMS Review Comments 3-18-25
This manuscript considers possible adverse health effects caused by radiation exposure during normal medical CT scans. The damage is reported to involve DNA damage, as observed from peripheral blood cells. This is an important topic, one that will be of interest to many readers. The research merits publication, but I would like to see some changes.
I am concerned about some generalizations made by the authors regarding radiation levels during radiographic medical examinations. For example, line 234: Given the very low doses administered in the course of a CT examination, only subtle biological effects can be expected.
The radiation dosage statement curious. For example, conventional chest x-rays typically have radiation exposures of around 0.02 mSV, with perhaps 1.5 mSV for a lumbar X-ray. Compare this to abdominal CT scans where the exposures may be 7 or 8 mSV. Of course, CT scan exposures are very low compared to fluoroscopy, but the statement “very low doses administered in the course of a CT scan” is a subjective opinion. Consider the statement in line 117: we divided the patient group into three subsets as described above (low, medium and high exposure group).
The dose issue is not addressed until near the end of the manuscript, in the methods section. I find this organization to be problematic. The presentation would be clearer if the methods section had the normal placement where it follows the introduction, and precedes the results section. I have seen man uscripts (and written some of them) where materials and methods section is placed at the end of the paper as an appendix. In this instance, placing the methods as a regular section makes the presentation end with a whimper rather than a bang. More importantly, the presentation of results lacks clarity because of the delay in describing the methods.
Some very minor comments:
Line 108: Figure 1. (a) Volcano plot highlighting genes differentially regulated after CT exposure. padj values
“padj” begins a new sentence without a capital letter.
The authors describe graphic plots as “volcano plots” (figure 1), “violin plots” (figure 2), “upset” plot (figure 3). Perhaps these could all just be called “plots” or “graphs”.
Author Response
Reviewer 2:
This manuscript considers possible adverse health effects caused by radiation exposure during normal medical CT scans. The damage is reported to involve DNA damage, as observed from peripheral blood cells. This is an important topic, one that will be of interest to many readers. The research merits publication, but I would like to see some changes.
I am concerned about some generalizations made by the authors regarding radiation levels during radiographic medical examinations. For example, line 234: Given the very low doses administered in the course of a CT examination, only subtle biological effects can be expected.
The radiation dosage statement curious. For example, conventional chest x-rays typically have radiation exposures of around 0.02 mSV, with perhaps 1.5 mSV for a lumbar X-ray. Compare this to abdominal CT scans where the exposures may be 7 or 8 mSV. Of course, CT scan exposures are very low compared to fluoroscopy, but the statement “very low doses administered in the course of a CT scan” is a subjective opinion. Consider the statement in line 117: we divided the patient group into three subsets as described above (low, medium and high exposure group).
We thank the reviewer for this important remark. The dose levels refer to the usual classification of low-dose (<100 mSv) and ultra-low-dose (<10 mSv) in radiobiology. This can indeed be confusing, as in diagnostic radiology CT with doses above 1mSv is considered a high-dose examination. We have adapted the passage accordingly. It now reads: “Given the rather low doses administered in the course of a CT examination in a radiobiological context, only subtle biological effects can be expected.”
The dose issue is not addressed until near the end of the manuscript, in the methods section. I find this organization to be problematic. The presentation would be clearer if the methods section had the normal placement where it follows the introduction, and precedes the results section. I have seen man uscripts (and written some of them) where materials and methods section is placed at the end of the paper as an appendix. In this instance, placing the methods as a regular section makes the presentation end with a whimper rather than a bang. More importantly, the presentation of results lacks clarity because of the delay in describing the methods.
We agree with the reviewer on this point. By placing the M&M section at the end, we followed the author guidelines of the journal.
Some very minor comments:
Line 108: Figure 1. (a) Volcano plot highlighting genes differentially regulated after CT exposure. padj values
“padj” begins a new sentence without a capital letter.
Thank you for the remark. We adjusted the manuscript accordingly.
The authors describe graphic plots as “volcano plots” (figure 1), “violin plots” (figure 2), “upset” plot (figure 3). Perhaps these could all just be called “plots” or “graphs”.
For a better understanding of the graphics, we would prefer an exact naming of each plot.
Round 2
Reviewer 2 Report
Comments and Suggestions for Authors
Thanks for considering my comments